# Rapid Distance-Based Outlier Detection via Sampling

**Mahito Sugiyama**[1]    **Karsten M. Borgwardt**[1,2]
[1]Machine Learning and Computational Biology Research Group, MPIs Tübingen, Germany
[2]Zentrum für Bioinformatik, Eberhard Karls Universität Tübingen, Germany
{mahito.sugiyama,karsten.borgwardt}@tuebingen.mpg.de

## Abstract

Distance-based approaches to outlier detection are popular in data mining, as they do not require to model the underlying probability distribution, which is particularly challenging for high-dimensional data. We present an empirical comparison of various approaches to distance-based outlier detection across a large number of datasets. We report the surprising observation that a simple, sampling-based scheme outperforms state-of-the-art techniques in terms of both efficiency and effectiveness. To better understand this phenomenon, we provide a theoretical analysis why the sampling-based approach outperforms alternative methods based on $k$-nearest neighbor search.

## 1 Introduction

An *outlier*, which is "an observation which deviates so much from other observations as to arouse suspicions that it was generated by a different mechanism" (by Hawkins [10]), appears in many real-life situations. Examples include intrusions in network traffic, credit card frauds, defective products in industry, and misdiagnosed patients. To discriminate such outliers from normal observations, machine learning and data mining have defined numerous outlier detection methods, for example, traditional model-based approaches using statistical tests, convex full layers, or changes of variances and more recent distance-based approaches using $k$-nearest neighbors [18], clusters [23], or densities [7] (for reviews, see [1, 13]).

We focus in this paper on the latter, the distance-based approaches, which define outliers as objects located far away from the remaining objects. More specifically, given a metric space $(\mathcal{M}, d)$, each object $\boldsymbol{x} \in \mathcal{M}$ receives a real-valued outlierness score $q(\boldsymbol{x})$ via a function $q : \mathcal{M} \rightarrow \mathbb{R}$; $q(\boldsymbol{x})$ depends on the distances between $\boldsymbol{x}$ and the other objects in the dataset. Then the top-$\kappa$ objects with maximum outlierness scores are reported to be outliers. To date, this approach has been successfully applied in various situations due to its flexibility, that is, it does not require to determine or to fit an underlying probability distribution, which is often difficult, in particular in high-dimensional settings. For example, LOF (Local Outlier Factor) [7] has become one of the most popular outlier detection methods, which measures the outlierness of each object by the difference of local densities between the object and its neighbors.

The main challenge, however, is its *scalability* since this approach potentially requires computation of all pairwise distances between objects in a dataset. This quadratic time complexity leads to runtime problems on massive datasets that emerge across application domains. To avoid this high computational cost, a number of techniques have already been proposed, which can be roughly divided into two strategies: *indexing* of objects such as tree-based structures [5] or projection-based structures [9] and *partial computation* of the pairwise distances to compute scores only for the top-$\kappa$ outliers, first introduced by Bay and Schwabacher [4] and improved in [6, 16]. Unfortunately, both strategies are nowadays not sufficient, as index structures are often not efficient enough for high-dimensional data [20] and the number of outliers often increases in direct proportion to the size of the dataset, which significantly deteriorates the efficiency of partial computation techniques.

Here we show that a surprisingly simple and rapid *sampling-based* outlier detection method outperforms state-of-the-art distance-based methods in terms of both efficiency and effectiveness by conducting an extensive empirical analysis. The proposed method behaves as follows: It takes a small set of samples from a given set of objects, followed by measuring the outlierness of each object by the distance from the object to its *nearest neighbor in the sample set*. Intuitively, the sample set is employed as a *telltale set*, that is, it serves as an indicator of outlierness, as outliers should be significantly different from *almost all* objects by definition, including the objects in the sample set. The time complexity is therefore linear in the number of objects, dimensions, and samples. In addition, this method can be implemented in a *one-pass* manner with constant space complexity as we only have to store the sample set, which is ideal for analyzing massive datasets.

This paper is organized as follows: In Section 2, we describe our experimental design for the empirical comparison of different outlier detection strategies. In Section 3, we review a number of state-of-the-art outlier detection methods which we used in our experiments, including our own proposal. We present experimental results in Section 4 and theoretically analyze them in Section 5.

## 2 Experimental Design

We present an extensive empirical analysis of state-of-the-art approaches for distance-based outlier detection and of our new approach, which are introduced in Section 3. They are evaluated in terms of both scalability and effectiveness on synthetic and real-world datasets. All parameters are set by referring the original literature or at popular values, which are also shown in Section 3. Note that these parameters have to be chosen by heuristics in distance-based approaches, while they still outperform other approaches such as statistical approaches [3].

**Environment.** We used Ubuntu version 12.04.3 with a single 2.6 GHz AMD Opteron CPU and 512 GB of memory. All C codes were compiled with `gcc` 4.6.3. All experiments were performed in the R environment, version 3.0.1.

**Evaluation criterion.** To evaluate the effectiveness of each method, we used the area under the precision-recall curve (AUPRC; equivalent to the average precision), which is a typical criterion to measure the success of outlier detection methods [1]. It takes values from 0 to 1 and 1 is the best score, and quantifies whether the algorithm is able to retrieve outliers correctly. These values were calculated by the R `ROCR` package.

**Datasets.** We collected 14 real-world datasets from the UCI machine learning repository [2], with a wide range of sizes and dimensions, whose properties are summarized in Table 1. Most of them have been intensively used in the outlier detection literature. In particular, `KDD1999` is one of the most popular benchmark datasets in outlier detection, which was originally used for the KDD Cup 1999. The task is to detect intrusions from network traffic data, and as in [22], objects whose attribute `logged_in` is positive were chosen as outliers. In every dataset, we first excluded all categorical attributes and missing values since some methods cannot handle categorical attributes. For all datasets except for `KDD1999`, we assume that objects from the smallest class are outliers, as they are originally designed for classification rather than outlier detection. Three datasets `Mfeat`, `Isolet`, and `Optdigits` were prepared exactly the same way as [17], where only two similar classes were used as inliers. All datasets were normalized beforehand, that is, in each dimension, the feature values were divided by their standard deviation [1, Chapter 12.10].

In addition, we generated two synthetic datasets (`Gaussian`) using exactly the same procedure as [14, 17], of which one is high-dimensional (1000 dimensions) and the other is large (10,000,000 objects). For each dataset, inliers (non-outliers) were generated from a Gaussian mixture model with five equally weighted processes, resulting in five clusters. The mean and the variance of each cluster was randomly set from the Gaussian distribution $N(0, 1)$, and 30 outliers were generated from a uniform distribution in the range from the minimum to the maximum values of inliers.

## 3 Methods for Outlier Detection

In the following, we will introduce the state-of-the-art methods in distance-based outlier detection, including our new sampling-based method. Every method is formalized as a scoring function $q : \mathcal{M} \to \mathbb{R}$ on a metric space $(\mathcal{M}, d)$, which assigns a real-valued outlierness score to each object $\boldsymbol{x}$

in a given set of objects $\mathcal{X}$. We denote by $n$ the number of objects in $\mathcal{X}$. If $\mathcal{X}$ is multivariate, the number of dimensions is denoted by $m$. The number of samples (sample size) is denoted by $s$.

## 3.1 The *k*th-nearest neighbor distance

Knorr and Ng [11, 12] were the first to formalize a distance-based outlier detection scheme, in which an object $\boldsymbol{x} \in \mathcal{X}$ is said to be a *DB*$(\alpha, \delta)$-*outlier* if $|\{\boldsymbol{x}' \in \mathcal{X} \mid d(\boldsymbol{x}, \boldsymbol{x}') > \delta\}| \geq \alpha n$, where $\alpha$ and $\delta$ with $\alpha, \delta \in \mathbb{R}$ and $0 \leq \alpha \leq 1$ are parameters specified by the user. This means that at least a fraction $\alpha$ of all objects have a distance from $x$ that is larger than $\delta$. This definition has mainly two significant drawbacks: the difficulty of determining the distance threshold $\delta$ in practice and the lack of a ranking of outliers. To overcome these drawbacks, Ramaswamy *et al.* [18] proposed to measure the outlierness by the *kth-nearest neighbor* (*k*th-NN) *distance*. The score $q_{k\text{thNN}}(\boldsymbol{x})$ of an object $\boldsymbol{x}$ is defined as

$$q_{k\text{thNN}}(\boldsymbol{x}) := d^k(\boldsymbol{x}; \mathcal{X}),$$

where $d^k(\boldsymbol{x}; \mathcal{X})$ is the distance between $\boldsymbol{x}$ and its $k$th-NN in $\mathcal{X}$. Notice that if we set $\alpha = (n-k)/n$, the set of Knorr and Ng's DB$(\alpha, \delta)$-outliers coincides with the set $\{\boldsymbol{x} \in \mathcal{X} \mid q_{k\text{thNN}}(\boldsymbol{x}) \geq \delta\}$. We employ $q_{k\text{thNN}}(\boldsymbol{x})$ as a baseline for distance-based methods in our comparison.

Since the naïve computation of scores $q_{k\text{thNN}}(\boldsymbol{x})$ for all $\boldsymbol{x}$ requires quadratic computational cost, a number of studies investigated speed-up techniques [4, 6, 16]. We used Bhaduri's algorithm (called iORCA) [6] and implemented it in C since it is the latest technique in this branch of research. It has a parameter $k$ to specify the $k$th-NN and an additional parameter $\kappa$ to retrieve the top-$\kappa$ objects with the largest outlierness scores. We set $k = 5$, which is a default setting used in the literature [4, 6, 15, 16], and set $\kappa$ to be twice the number of outliers for each dataset. Note that in practice we usually do not know the exact number of outliers and have to set $\kappa$ large enough.

## 3.2 Iterative sampling

Wu and Jermaine [21] proposed a sampling-based approach to efficiently approximate the $k$th-NN distance score $q_{k\text{thNN}}$. For each object $\boldsymbol{x} \in \mathcal{X}$, define

$$q_{k\text{thSp}}(\boldsymbol{x}) := d^k(\boldsymbol{x}; S_{\boldsymbol{x}}(\mathcal{X})),$$

where $S_{\boldsymbol{x}}(\mathcal{X})$ is a subset of $\mathcal{X}$, which is randomly and iteratively sampled for each object $\boldsymbol{x}$. In addition, they introduced a random variable $N = |\mathcal{O} \cap \mathcal{O}'|$ with two sets of top-$\kappa$ outliers $\mathcal{O}$ and $\mathcal{O}'$ with respect to $q_{k\text{thNN}}$ and $q_{k\text{thSp}}$, and analyzed its expectation $\mathrm{E}(N)$ and the variance $\mathrm{Var}(N)$. The time complexity is $\Theta(nms)$. We implemented this method in C and set $k = 5$ and the sample size $s = 20$ unless stated otherwise.

## 3.3 One-time sampling (our proposal)

Here we present a new sampling-based method. We randomly and independently sample a subset $S(\mathcal{X}) \subset \mathcal{X}$ only once and define

$$q_{\text{Sp}}(\boldsymbol{x}) := \min_{\boldsymbol{x}' \in S(\mathcal{X})} d(\boldsymbol{x}, \boldsymbol{x}')$$

for each object $\boldsymbol{x} \in \mathcal{X}$. Although this definition is closely related to Wu and Jermaine's method $q_{k\text{thSp}}$ in the case of $k = 1$, our method performs sampling only once while their method performs sampling for each object. We empirically show that this leads to significant differences in accuracy in outlier detection (see Section 4). We also theoretically analyze this phenomenon to get a better understanding of its cause (see Section 5). The time complexity is $\Theta(nms)$ and the space complexity is $\Theta(ms)$ using the number of samples $s$, as this score can be obtained in a one-pass manner. We implemented this method in C. We set $s = 20$ for the comparison with other methods.

## 3.4 Isolation forest

Liu *et al.* [15] proposed a random forest-like method, called *isolation forest*. It uses random recursive partitions of objects, which are assumed to be $m$-dimensional vectors, and hence is also based on the concept of proximity. From a given set $\mathcal{X}$, we construct an *iTree* in the following manner. First a sample set $S(\mathcal{X}) \subset \mathcal{X}$ is chosen. Then this sample set is partitioned into two non-empty subsets

$S(\mathcal{X})_{\mathrm{L}}$ and $S(\mathcal{X})_{\mathrm{R}}$ such that $S(\mathcal{X})_{\mathrm{L}} = \{\, \boldsymbol{x} \in S(\mathcal{X}) \mid x_q < v \,\}$ and $S(\mathcal{X})_{\mathrm{R}} = S(\mathcal{X}) \backslash S(\mathcal{X})_{\mathrm{L}}$, where $v$ and $q$ are randomly chosen. This process is recursively applied to each subset until it becomes a singleton, resulting in a proper binary tree such that the number of nodes is $2s - 1$. The outlierness of an object $\boldsymbol{x}$ is measured by the *path length* $h(\boldsymbol{x})$ on the tree, and the score is normalized and averaged on $t$ *i*Trees. Finally, the outlierness score $q_{\mathrm{tree}}(\boldsymbol{x})$ is defined as

$$q_{\mathrm{tree}}(\boldsymbol{x}) := 2^{-\overline{h(\boldsymbol{x})}/c(s)},$$

where $\overline{h(\boldsymbol{x})}$ is the average of $h(\boldsymbol{x})$ on $t$ *i*Trees and $c(s)$ is defined as $c(s) := 2H(s-1) - 2(s-1)/n$, where $H$ denotes the harmonic number. The overall average and worst case time complexities are $O((s+n)t \log s)$ and $O((s+n)ts)$. We used the official R `IsolationForest` package[1], whose core process is implemented in C. We set $t = 100$ and $s = 256$, which is the same setting as in [15].

### 3.5 Local outlier factor (LOF)

While LOF [7] is often referred to as not distance-based but *density-based*, we still include this method as it is also based on pairwise distances and is known to be a prominent outlier detection method. Let $N^k(\boldsymbol{x})$ be the set of $k$-nearest neighbors of $\boldsymbol{x}$. The local reachability density of $\boldsymbol{x}$ is defined as $\rho(\boldsymbol{x}) := |N^k(\boldsymbol{x})| \left( \sum_{\boldsymbol{x}' \in N^k(\boldsymbol{x})} \max\{\, d^k(\boldsymbol{x}', \mathcal{X}), d(\boldsymbol{x}, \boldsymbol{x}') \,\} \right)^{-1}$. Then the *local outlier factor* (LOF) $q_{\mathrm{LOF}}(\boldsymbol{x})$ is defined as the ratio of the local reachability density of $\boldsymbol{x}$ and the average of the local reachability densities of its $k$-nearest neighbors, that is,

$$q_{\mathrm{LOF}}(\boldsymbol{x}) := \left( |N^k(\boldsymbol{x})|^{-1} \sum_{\boldsymbol{x}' \in N^k(\boldsymbol{x})} \rho(\boldsymbol{x}') \right) \rho(\boldsymbol{x})^{-1}.$$

The time complexity is $O(n^2 m)$, which is known to be the main disadvantage of this method. We implemented this method in C and used the commonly used setting $k = 10$.

### 3.6 Angle-based outlier factor (ABOF)

Kriegel *et al.* [14] proposed to use *angles* instead of distances to measure outlierness. Let $c(\boldsymbol{x}, \boldsymbol{x}')$ be the *similarity* between vectors $\boldsymbol{x}$ and $\boldsymbol{x}'$, for example, the cosine similarity. Then $c(\boldsymbol{y} - \boldsymbol{x}, \boldsymbol{y}' - \boldsymbol{x})$ should be correlated with the angle of two vectors $\boldsymbol{y}$ and $\boldsymbol{y}'$ with respect to the the coordinate origin $\boldsymbol{x}$. The insight of Kriegel *et al.* is that if $\boldsymbol{x}$ is an outlier, the *variance of angles* between pairs of the remaining objects becomes small. Formally, for an object $\boldsymbol{x} \in \mathcal{X}$ define

$$q_{\mathrm{ABOF}}(\boldsymbol{x}) := \mathrm{Var}_{\boldsymbol{y}, \boldsymbol{y}' \in \mathcal{X}}\, c(\boldsymbol{y} - \boldsymbol{x}, \boldsymbol{y}' - \boldsymbol{x}).$$

Note that the smaller $q_{\mathrm{ABOF}}(\boldsymbol{x})$, the more likely is $\boldsymbol{x}$ to be an outlier, which is in contrast to the other methods. This method was originally introduced to overcome the "curse of dimensionality" in high-dimensional data. However, recently Zimek *et al.* [24] showed that distance-based methods such as LOF also work if attributes carry relevant information for outliers. We include several high-dimensional datasets in experiments and check whether distance-based methods work effectively.

Although this method is attractive as it is *parameter-free*, the computational cost is cubic in $n$. Thus we use its near-linear approximation algorithm proposed by Pham and Pagh [17]. Their algorithm, called FastVOA, estimates the first and the second moments of the variance $\mathrm{Var}_{\boldsymbol{y}, \boldsymbol{y}' \in \mathcal{X}}\, c(\boldsymbol{y} - \boldsymbol{x}, \boldsymbol{y}' - \boldsymbol{x})$ independently using two techniques: random projections and AMS sketches. The latter is a randomized technique to estimate the second frequency moment of a data stream. The resulting time complexity is $O(tn(m + \log n + c_1 c_2))$, where $t$ is the number of hyperplanes for random projections and $c_1, c_2$ are the number of repetitions for AMS sketches. We implemented this algorithm in C. We set $t = \log n$, $c_1 = 1600$, and $c_2 = 10$ as they are shown to be empirically sufficient in [17].

### 3.7 One-class SVM

The *One-class SVM*, introduced by Schölkopf *et al.* [19], classifies objects into inliers and outliers by introducing a hyperplane between them. This classification can be turned into a ranking of outlierness by considering the signed distance to the separating hyperplane. That is, the further an object is located in the outlier half space, the more likely it is to be a true outlier. Let $\mathcal{X} = \{\boldsymbol{x}_1, \ldots, \boldsymbol{x}_n\}$. Formally, the score of a vector $\boldsymbol{x}$ with a feature map $\Phi$ is defined as

$$q_{\mathrm{SVM}}(\boldsymbol{x}) := \rho - (\boldsymbol{w} \cdot \Phi(\boldsymbol{x})), \tag{1}$$

Table 1: Summary of datasets. Gaussian is synthetic (marked by *) and the other datasets are collected from the UCI repository ($n$ = number of objects, $m$ = number of dimensions).

| | $n$ | # of outliers | $m$ |
|---|---|---|---|
| Ionosphere | 351 | 126 | 34 |
| Arrhythmia | 452 | 207 | 274 |
| Wdbc | 569 | 212 | 30 |
| Mfeat | 600 | 200 | 649 |
| Isolet | 960 | 240 | 617 |
| Pima | 768 | 268 | 8 |
| Gaussian* | 1000 | 30 | 1000 |
| Optdigits | 1688 | 554 | 64 |
| Spambase | 4601 | 1813 | 57 |
| Statlog | 6435 | 626 | 36 |
| Skin | 245057 | 50859 | 3 |
| Pamap2 | 373161 | 125953 | 51 |
| Covtype | 286048 | 2747 | 10 |
| Kdd1999 | 4898431 | 703067 | 6 |
| Record | 5734488 | 20887 | 7 |
| Gaussian* | 10000000 | 30 | 20 |

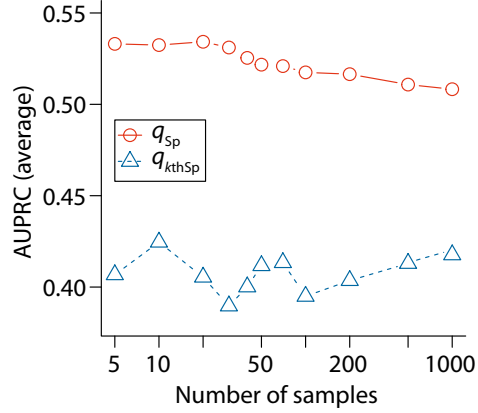

Figure 1: Average of area under the precision-recall curves (AUPRCs) over all datasets with respect to changes in number of samples $s$ for $q_{\mathrm{Sp}}$ (one-time sampling; our proposal) and $q_{k\mathrm{thSp}}$ (iterative sampling by Wu and Jermaine [21]). Note that the $x$-axis has logarithmic scale.

where the weight vector $\boldsymbol{w}$ and the offset $\rho$ are optimized by the following quadratic program:

$$\min_{\boldsymbol{w}\in\mathcal{F},\,\boldsymbol{\xi}\in\mathbb{R}^n,\,\rho\in\mathbb{R}} \frac{1}{2}\|\boldsymbol{w}\|^2 + \frac{1}{\nu n}\sum_{i=1}^n \xi_i - \rho \quad \text{subject to } (\boldsymbol{w}\cdot\Phi(\boldsymbol{x}_i)) \geq \rho - \xi_i,\ \xi_i \geq 0$$

with a regularization parameter $\nu$. The term $\boldsymbol{w}\cdot\Phi(\boldsymbol{x})$ in equation (1) can be replaced with $\sum_{i=1}^n \alpha_i k(\boldsymbol{x}_i, \boldsymbol{x})$ using a kernel function $k$, where $\boldsymbol{\alpha} = (\alpha_1, \ldots, \alpha_n)$ is used in the dual problem.

We tried ten different values of $\nu$ from 0 to 1 and picked up the one maximizing the margin between negative and positive scores. We used a Gaussian RBF kernel and set its parameter $\sigma$ by the popular heuristics [8]. The R `kernlab` package was used, whose core process is implemented in C.

# 4 Experimental Results

## 4.1 Sensitivity in sampling size and sampling scheme

We first analyze the parameter sensitivity of our method $q_{\mathrm{Sp}}$ with respect to changes in the sample size $s$. In addition, for each sample size we compare our $q_{\mathrm{Sp}}$ (one-time sampling) to Wu and Jermaine's $q_{k\mathrm{thSp}}$ (iterative sampling). We set $k = 1$ in $q_{k\mathrm{thSp}}$, hence the only difference between them was the sampling scheme. Each method was applied to each dataset listed in Table 1 and the average of AUPRCs (area under the precision-recall curves) in 10 trials were obtained, and these were again averaged over all datasets. These scores with varying sample sizes are plotted in Figure 1.

Our method shows robust performance over all sample sizes from 5 to 1000 and the average AUPRC varies by less than 2%. Interestingly, the score is maximized at a rather small sample size ($s = 20$) and monotonically (slightly) decreases with increasing sample size. Moreover, for every sample size, the one-time sampling $q_{\mathrm{Sp}}$ significantly outperforms the iterative sampling $q_{k\mathrm{thSp}}$ (Wilcoxon signed-rank test, $\alpha = 0.05$). We checked that this behavior is independent from dataset size.

## 4.2 Scalability and effectiveness

Next we evaluate the scalability and effectiveness of the approaches introduced in Section 3 by systematically applying them to every dataset. Results of running time and AUPRCs are shown in Table 2 and Table 3, respectively. As we can see, our method $q_{\mathrm{Sp}}$ is the fastest among all methods; it can score more than five million objects within a few seconds. Although the time complexity of Wu and Jermaine's $q_{k\mathrm{thSp}}$ is the same as $q_{\mathrm{Sp}}$, our method is empirically much faster, especially in large datasets. The different costs of two processes, sampling once and performing nearest neighbor

Table 2: Running time (in seconds). Averages in 10 trials are shown in four probabilistic methods $q_{k\text{thSp}}$, $q_{\text{Sp}}$, $q_{\text{tree}}$, and $q_{\text{ABOF}}$. "—" means that computation did not completed within 2 months.

| | $q_{k\text{thNN}}$ | $q_{k\text{thSp}}$ | $q_{\text{Sp}}$ | $q_{\text{tree}}$ | $q_{\text{LOF}}$ | $q_{\text{ABOF}}$ | $q_{\text{SVM}}$ |
|---|---|---|---|---|---|---|---|
| Ionosphere | $2.00\times10^{-2}$ | $9.60\times10^{-3}$ | $\mathbf{8.00\times10^{-4}}$ | $6.25\times10^{-1}$ | $2.40\times10^{-2}$ | $4.72$ | $6.80\times10^{-2}$ |
| Arrhythmia | $2.56\times10^{-1}$ | $2.72\times10^{-2}$ | $\mathbf{1.52\times10^{-2}}$ | $2.72$ | $2.04\times10^{-1}$ | $6.19$ | $3.88\times10^{-1}$ |
| Wdbc | $7.20\times10^{-2}$ | $1.60\times10^{-2}$ | $\mathbf{2.00\times10^{-3}}$ | $7.32\times10^{-1}$ | $6.80\times10^{-2}$ | $7.86$ | $9.20\times10^{-2}$ |
| Mfeat | $1.04$ | $6.00\times10^{-2}$ | $\mathbf{4.80\times10^{-2}}$ | $8.69$ | $1.02$ | $8.26$ | $1.90$ |
| Isolet | $4.27$ | $8.68\times10^{-2}$ | $\mathbf{8.37\times10^{-2}}$ | $9.71$ | $4.61$ | $1.38\times10^{1}$ | $3.60$ |
| Pima | $4.00\times10^{-2}$ | $2.04\times10^{-2}$ | $\mathbf{4.00\times10^{-4}}$ | $3.14\times10^{-1}$ | $9.20\times10^{-2}$ | $1.07\times10^{1}$ | $9.60\times10^{-2}$ |
| Gaussian | $4.18$ | $2.13\times10^{-1}$ | $\mathbf{1.54\times10^{-1}}$ | $2.10\times10^{1}$ | $2.61\times10^{1}$ | $1.46\times10^{1}$ | $7.77$ |
| Optdigits | $1.04$ | $7.48\times10^{-2}$ | $\mathbf{1.48\times10^{-2}}$ | $8.65\times10^{-1}$ | $1.46$ | $2.41\times10^{1}$ | $1.14$ |
| Spambase | $9.51$ | $7.26\times10^{-1}$ | $\mathbf{3.68\times10^{-2}}$ | $1.02$ | $1.14\times10^{1}$ | $7.75\times10^{1}$ | $8.77$ |
| Statlog | $6.99$ | $2.03\times10^{-1}$ | $\mathbf{2.80\times10^{-2}}$ | $9.35\times10^{-1}$ | $1.68\times10^{1}$ | $1.07\times10^{2}$ | $1.39\times10^{1}$ |
| Skin | $6.82\times10^{3}$ | $2.12\times10^{1}$ | $\mathbf{9.72\times10^{-2}}$ | $3.04$ | $1.38\times10^{4}$ | $7.33\times10^{3}$ | $9.44\times10^{3}$ |
| Pamap2 | $9.05\times10^{4}$ | $3.27\times10^{1}$ | $\mathbf{2.73}$ | $1.20\times10^{1}$ | $1.37\times10^{5}$ | $1.71\times10^{4}$ | $8.37\times10^{4}$ |
| Covtype | $6.87\times10^{2}$ | $2.16\times10^{1}$ | $\mathbf{2.83\times10^{-1}}$ | $6.15$ | $3.67\times10^{4}$ | $1.13\times10^{4}$ | $1.69\times10^{4}$ |
| Kdd1999 | $2.68\times10^{6}$ | $4.40\times10^{2}$ | $\mathbf{3.46}$ | $4.78\times10^{1}$ | — | $2.40\times10^{5}$ | — |
| Record | $3.62\times10^{6}$ | $9.58\times10^{2}$ | $\mathbf{4.11}$ | $8.84\times10^{1}$ | — | $1.07\times10^{6}$ | — |
| Gaussian | $3.37\times10^{3}$ | $1.73\times10^{3}$ | $\mathbf{2.13\times10^{1}}$ | $3.26\times10^{2}$ | — | $1.47\times10^{6}$ | — |

Table 3: Area under the precision-recall curve (AUPRC). Averages±SEMs in 10 trials are shown in four probabilistic methods. Best scores are denoted in Bold. Note that the root mean square deviation (RMSD) rewards methods that are always close to the best result on each dataset.

| | $q_{k\text{thNN}}$ | $q_{k\text{thSp}}$ | $q_{\text{Sp}}$ | $q_{\text{tree}}$ | $q_{\text{LOF}}$ | $q_{\text{ABOF}}$ | $q_{\text{SVM}}$ |
|---|---|---|---|---|---|---|---|
| Ionosphere | $\mathbf{0.931}$ | $0.762\pm0.007$ | $0.899\pm0.032$ | $0.871\pm0.002$ | $0.864$ | $0.740\pm0.022$ | $0.794$ |
| Arrhythmia | $0.701$ | $0.674\pm0.008$ | $\mathbf{0.711}\pm0.005$ | $0.681\pm0.004$ | $0.673$ | $0.697\pm0.005$ | $0.707$ |
| Wdbc | $0.607$ | $0.226\pm0.001$ | $\mathbf{0.667}\pm0.036$ | $0.595\pm0.018$ | $0.428$ | $0.490\pm0.014$ | $0.556$ |
| Mfeat | $0.217$ | $0.293\pm0.002$ | $0.245\pm0.031$ | $0.270\pm0.009$ | $\mathbf{0.369}$ | $0.211\pm0.003$ | $0.257$ |
| Isolet | $0.380$ | $0.175\pm0.001$ | $\mathbf{0.535}\pm0.138$ | $0.328\pm0.011$ | $0.274$ | $0.520\pm0.034$ | $0.439$ |
| Pima | $0.519$ | $\mathbf{0.608}\pm0.007$ | $0.512\pm0.010$ | $0.441\pm0.003$ | $0.406$ | $0.461\pm0.008$ | $0.461$ |
| Gaussian | $\mathbf{1.000}$ | $\mathbf{1.000}\pm0.000$ | $\mathbf{1.000}\pm0.000$ | $0.934\pm0.036$ | $0.904$ | $0.994\pm0.005$ | $\mathbf{1.000}$ |
| Optdigits | $0.204$ | $0.319\pm0.001$ | $0.233\pm0.021$ | $0.295\pm0.010$ | $\mathbf{0.361}$ | $0.255\pm0.006$ | $0.266$ |
| Spambase | $0.395$ | $0.418\pm0.001$ | $\mathbf{0.422}\pm0.011$ | $0.419\pm0.011$ | $0.354$ | $0.398\pm0.002$ | $0.399$ |
| Statlog | $0.057$ | $0.058\pm0.000$ | $0.082\pm0.008$ | $0.060\pm0.002$ | $\mathbf{0.093}$ | $0.054\pm0.000$ | $0.056$ |
| Skin | $0.195$ | $0.146\pm0.000$ | $\mathbf{0.353}\pm0.058$ | $0.242\pm0.003$ | $0.130$ | $0.258\pm0.006$ | $0.213$ |
| Pamap2 | $0.249$ | $0.328\pm0.000$ | $0.268\pm0.009$ | $0.252\pm0.001$ | $\mathbf{0.338}$ | $0.231\pm0.002$ | $0.235$ |
| Covtype | $0.016$ | $0.058\pm0.001$ | $0.075\pm0.034$ | $0.017\pm0.001$ | $0.010$ | $0.087\pm0.005$ | $\mathbf{0.095}$ |
| Kdd1999 | $\mathbf{0.768}$ | $0.081\pm0.000$ | $0.611\pm0.098$ | $0.389\pm0.007$ | — | $0.539\pm0.020$ | — |
| Record | $0.002$ | $0.411\pm0.000$ | $0.933\pm0.013$ | $\mathbf{0.976}\pm0.004$ | — | $0.658\pm0.106$ | — |
| Gaussian | $\mathbf{1.000}$ | $0.999\pm0.000$ | $\mathbf{1.000}\pm0.000$ | $0.890\pm0.022$ | — | $0.893\pm0.003$ | — |
| Average | $0.453$ | $0.410$ | $\mathbf{0.534}$ | $0.479$ | $0.400$ | $0.468$ | $0.421$ |
| Avg.Rank | $3.750$ | $3.875$ | $\mathbf{2.188}$ | $3.875$ | $4.538$ | $4.563$ | $4.000$ |
| RMSD | $0.259$ | $0.274$ | $\mathbf{0.068}$ | $0.133$ | $0.152$ | $0.140$ | $0.094$ |

search versus re-sampling per object and performing $k$th-NN search, causes this difference. The baseline $q_{k\text{thNN}}$ shows acceptable runtimes for large data only if the number of outliers is small.

In terms of effectiveness, $q_{\text{Sp}}$ shows the best performance on seven out of sixteen datasets including the high-dimensional datasets, resulting in the best average AUPRC score, which is significantly higher than every single method except for $q_{\text{LOF}}$ (Wilcoxon signed-rank test, $\alpha = 0.05$). The method $q_{\text{Sp}}$ also shows the best performance in terms of the average rank and RMSDs (root mean square deviations) to the best result on each dataset. Moreover, $q_{\text{Sp}}$ is inferior to the baseline $q_{k\text{thNN}}$ only on three datasets. It is interesting that $q_{\text{tree}}$, which also uses one-time sampling like our method, shows better performance than exhaustive methods on average. In contrast, $q_{k\text{thSp}}$ with iterative sampling is worst in terms of RMSD among all methods.

Based on these observations we can conclude that (1) small sample sizes lead to the maximum average precision for $q_{\text{Sp}}$; (2) one-time sampling leads to better results than iterative sampling; (3) one-time sampling leads to better results than exhaustive methods and is also much faster.

# 5 Theoretical Analysis

To understand why our new one-time sampling method $q_{\mathrm{Sp}}$ shows better performance than the other methods, we present a theoretical analysis to get answers to the following four questions: (1) What is the probability that $q_{\mathrm{Sp}}$ will correctly detect outliers? (2) Why do small sample sizes lead to better results in $q_{\mathrm{Sp}}$? (3) Why is $q_{\mathrm{Sp}}$ superior to $q_{k\mathrm{thSp}}$? (4) Why is $q_{\mathrm{Sp}}$ superior to $q_{k\mathrm{thNN}}$? Here we use the notion of Knorr and Ng's DB$(\alpha, \delta)$-outliers [11, 12] and denote the set of DB$(\alpha, \delta)$-outliers by $\mathcal{X}(\alpha; \delta)$, that is, an object $\boldsymbol{x} \in \mathcal{X}(\alpha; \delta)$ if $|\{\ \boldsymbol{x}' \in \mathcal{X} \mid d(\boldsymbol{x}, \boldsymbol{x}') > \delta\ \}| \geq \alpha n$ holds. We also define $\overline{\mathcal{X}}(\alpha; \delta) = \mathcal{X} \setminus \mathcal{X}(\alpha; \delta)$ and, for simplicity, we call an element in $\mathcal{X}(\alpha; \delta)$ an *outlier* and that in $\overline{\mathcal{X}}(\alpha; \delta)$ an *inlier* unless otherwise noted. Our method requires as input only the sample size $s$ in practice, whereas the parameters $\delta$ and $\alpha$ are used only in our theoretical analysis. In the following, we always assume that $s \ll n$, hence the sampling process is treated as with replacement.

**Probabilistic analysis of $q_{\mathrm{Sp}}$.** First we introduce a *partition* of inliers into subsets (clusters) using the threshold $\delta$. A $\delta$-*partition* $\boldsymbol{\mathcal{P}}_\delta$ of $\overline{\mathcal{X}}(\alpha; \delta)$ is defined as a set of non-empty disjoint subsets of $\overline{\mathcal{X}}(\alpha; \delta)$ such that each element (cluster) $\mathcal{C} \in \boldsymbol{\mathcal{P}}_\delta$ satisfies $\max_{\boldsymbol{x}, \boldsymbol{x}' \in \mathcal{C}} d(\boldsymbol{x}, \boldsymbol{x}') < \delta$ and $\bigcup_{\mathcal{C} \in \boldsymbol{\mathcal{P}}_\delta} \mathcal{C} = \overline{\mathcal{X}}(\alpha; \delta)$. Then if we focus on a cluster $\mathcal{C} \in \boldsymbol{\mathcal{P}}_\delta$, the probability of discriminating an outlier from inliers contained in $\mathcal{C}$ can be *bounded from below*. Remember that $s$ is the number of samples.

**Theorem 1** *For an outlier $\boldsymbol{x} \in \mathcal{X}(\alpha; \delta)$ and a cluster $\mathcal{C} \in \boldsymbol{\mathcal{P}}_\delta$, we have*

$$\Pr\big(\,\forall \boldsymbol{x}' \in \mathcal{C},\ q_{\mathrm{Sp}}(\boldsymbol{x}) > q_{\mathrm{Sp}}(\boldsymbol{x}')\,\big) \geq \alpha^s (1 - \beta^s) \quad \text{with } \beta = (n - |\mathcal{C}|)/n. \tag{2}$$

*Proof.* We have the probability $\Pr(q_{\mathrm{Sp}}(\boldsymbol{x}) > \delta) = \alpha^s$ from the definition of outliers. Moreover, if at least one object is sampled from the cluster $\mathcal{C}$, $q_{\mathrm{Sp}}(\boldsymbol{x}') < \delta$ holds for all $\boldsymbol{x}' \in \mathcal{C}$. Thus $\Pr(\forall \boldsymbol{x}' \in \mathcal{C},\ q_{\mathrm{Sp}}(\boldsymbol{x}') < \delta) = 1 - \beta^s$. Inequality (2) therefore follows. ∎

For instance, if we assume that 5% of our data are outliers and fix $\alpha$ to be 0.95, we have (maximum $\delta$, mean of $\beta$) = $(10.51, 0.50)$, $(44.25, 2.23 \times 10^{-3})$, $(10.93, 0.67)$, $(37.10, 0.75)$, and $(36.37, 0.80)$ on our first five datasets from Table 1 to achieve this 5% rate of outliers. These $\beta$ were obtained by greedily searching each cluster in $\boldsymbol{\mathcal{P}}_\delta$ under $\alpha = 0.95$ and the respective maximum $\delta$.

Next we consider the task of correctly discriminating an outlier from *all* inliers. This can be achieved if for each cluster $\mathcal{C} \in \boldsymbol{\mathcal{P}}_\delta$ at least one object $\boldsymbol{x} \in \mathcal{C}$ is chosen in the sampling process. Thus the lower bound can be directly derived using the multinomial distribution as follows.

**Theorem 2** *Let $\boldsymbol{\mathcal{P}}_\delta = \{\mathcal{C}_1, \ldots, \mathcal{C}_l\}$ with $l$ clusters and $p_i = |\mathcal{C}_i| / n$ for each $i \in \{1, \ldots, l\}$. For every outlier $\boldsymbol{x} \in \mathcal{X}(\alpha; \delta)$ and the sample size $s \geq l$, we have*

$$\Pr\big(\,\forall \boldsymbol{x}' \in \overline{\mathcal{X}}(\alpha; \delta),\ q_{\mathrm{Sp}}(\boldsymbol{x}) > q_{\mathrm{Sp}}(\boldsymbol{x}')\,\big) \geq \alpha^s \sum_{\forall i; s_i \geqslant 0} f(s_1, \ldots, s_l; s, p_1, \ldots, p_l),$$

*where $f$ is the probability mass function of the multinomial distribution defined as*

$$f(s_1, \ldots, s_l; s, p_1, \ldots, p_l) := (s! / \textstyle\prod_{i=1}^l s_i!) \prod_{i=1}^l p_i^{s_i} \quad \text{with } \textstyle\sum_{i=1}^l s_i = s.$$

Furthermore, let $\mathcal{I}(\alpha; \delta)$ be a subset of $\overline{\mathcal{X}}(\alpha; \delta)$ such that $\min_{\boldsymbol{x}' \in \mathcal{I}(\alpha; \delta)} d(\boldsymbol{x}, \boldsymbol{x}') > \delta$ for every outlier $\boldsymbol{x} \in \mathcal{X}(\alpha; \delta)$ and assume that $\boldsymbol{\mathcal{P}}_\delta$ is a $\delta$-partition of $\mathcal{I}(\alpha; \delta)$ instead of all inliers $\overline{\mathcal{X}}(\alpha; \delta)$. If $S(\mathcal{X}) \subseteq \mathcal{I}(\alpha; \delta)$ and at least one object is sampled from each cluster $\mathcal{C} \in \boldsymbol{\mathcal{P}}_\delta$, $q_{\mathrm{Sp}}(\boldsymbol{x}) > q_{\mathrm{Sp}}(\boldsymbol{x}')$ holds for all pairs of an outlier $\boldsymbol{x}$ and an inlier $\boldsymbol{x}'$.

**Theorem 3** *Let $\boldsymbol{\mathcal{P}}_\delta = \{\mathcal{C}_1, \ldots, \mathcal{C}_l\}$ be a $\delta$-partition of $\mathcal{I}(\alpha; \delta)$ and $\gamma = |\mathcal{I}(\alpha; \delta)| / n$, and assume that $p_i = |\mathcal{C}_i| / |\mathcal{I}(\alpha; \delta)|$ for each $i \in \{1, \ldots, l\}$. For every $s \geq l$,*

$$\Pr\big(\,\forall \boldsymbol{x} \in \mathcal{X}(\alpha; \delta), \forall \boldsymbol{x}' \in \overline{\mathcal{X}}(\alpha; \delta),\ q_{\mathrm{Sp}}(\boldsymbol{x}) > q_{\mathrm{Sp}}(\boldsymbol{x}')\,\big) \geq \gamma^s \sum_{\forall i; s_i \geqslant 0} f(s_1, \ldots, s_l; s, p_1, \ldots, p_l).$$

From the fact that this theorem holds for *any* $\delta$-partition, we automatically have the maximum lower bound over all possible $\delta$-partitions.

**Corollary 1** *Let $\varphi(s) = \sum_{\forall i; s_i \geqslant 0} f(s_1, \ldots, s_l; s, p_1, \ldots, p_l)$ given in Theorem 3. We have*

$$\Pr\big(\,\forall \boldsymbol{x} \in \mathcal{X}(\alpha; \delta), \forall \boldsymbol{x}' \in \overline{\mathcal{X}}(\alpha; \delta),\ q_{\mathrm{Sp}}(\boldsymbol{x}) > q_{\mathrm{Sp}}(\boldsymbol{x}')\,\big) \geq \gamma^s \max_{\boldsymbol{\mathcal{P}}_\delta} \varphi(s). \tag{3}$$

Let $B(\gamma; \delta)$ be the right-hand side of Inequality (3) above. This bound is maximized for equally sized clusters when $l$ is fixed and it shows high probability for large $\gamma$. For example if $\gamma = 0.99$, we have $(l, \text{optimal } s, B(\gamma; \delta)) = (2, 7, 0.918), (3, 12, 0.866)$, and $(4, 17, 0.818)$. It is notable that the bound $B(\gamma; \delta)$ is *independent* of the actual number of outliers and inliers, which is a desirable property when analyzing large datasets. Although it is dependent on the number of clusters $l$, the best (minimum) $l$ which maximizes $B(\gamma; \delta)$ with the simplest clustering is implicitly chosen in $q_{\text{Sp}}$.

**Theoretical support for small sample sizes.** Let $g(s) = \alpha^s(1 - \beta^s)$, which is the right-hand side of Inequality (2). From the differentiation $dg/ds$, we can see that this function is maximized at

$$s = \log_\beta \big( \log \alpha / (\log \alpha + \log \beta) \big),$$

with the natural assumption $0 < \beta < \alpha < 1$ and this optimal sample size $s$ is small for large $\alpha$ and small $\beta$, for example, $s = 6$ for $(\alpha, \beta) = (0.99, 0.5)$ and $s = 24$ for $(\alpha, \beta) = (0.999, 0.8)$. Moreover, as we already saw above the bound $B(\gamma; \delta)$ is also maximized at such small sample sizes for large $\gamma$. This could be the reason why $q_{\text{Sp}}$ works well for small sample sizes, as these are common values for $\alpha$, $\beta$, and $\gamma$ in outlier detection.

**Comparison with $q_{k\text{thSp}}$.** Define $Z(\boldsymbol{x}, \boldsymbol{x}') := \Pr(q_{k\text{thSp}}(\boldsymbol{x}) > q_{k\text{thSp}}(\boldsymbol{x}'))$ for the iterative sampling method $q_{k\text{thSp}}$. Since we repeat sampling for each object in $q_{k\text{thSp}}$, probability $Z(\boldsymbol{x}, \boldsymbol{x}')$ for each $\boldsymbol{x}' \in \overline{\mathcal{X}}(\alpha; \delta)$ is *independent* with respect to a fixed $\boldsymbol{x} \in \mathcal{X}(\alpha; \delta)$. We therefore have

$$\Pr\big( \forall \boldsymbol{x} \in \mathcal{X}(\alpha; \delta), \forall \boldsymbol{x}' \in \overline{\mathcal{X}}(\alpha; \delta), q_{k\text{thSp}}(\boldsymbol{x}) > q_{k\text{thSp}}(\boldsymbol{x}') \big) \le \min_{\boldsymbol{x} \in \mathcal{X}(\alpha; \delta)} \prod_{\boldsymbol{x}' \in \overline{\mathcal{X}}(\alpha; \delta)} Z(\boldsymbol{x}, \boldsymbol{x}').$$

Although $Z(\boldsymbol{x}, \boldsymbol{x}')$ is typically close to 1 in outlier detection, the overall probability rapidly decreases if $n$ is large. Thus the performance suffers on large datasets. In contrast, our one-time sampling $q_{\text{Sp}}$ does not have independence, resulting in our results (Theorem 1, 2, 3, and Corollary 1) instead of this upper bound, which often lead to higher probability. This fact might be the reason why $q_{k\text{thSp}}$ empirically performs significantly worse than $q_{\text{S}p}$ and shows the worst RMSD.

**Comparison with $q_{k\text{thNN}}$.** Finally, let us consider the situation in which there exists the set of "true" outliers $\mathcal{O} \subset \mathcal{X}$ given by an oracle. Let $\Lambda = \{k \in \mathbb{N} \mid q_{k\text{thNN}}(\boldsymbol{x}) > q_{k\text{thNN}}(\boldsymbol{x}')$ for all $\boldsymbol{x} \in \mathcal{O}$ and $\boldsymbol{x}' \in \mathcal{X} \setminus \mathcal{O}\}$, the set of $k$s with which we can detect all outliers, and assume that $\Lambda \neq \emptyset$. Then

$$\Pr\big( \forall \boldsymbol{x} \in \mathcal{O}, \forall \boldsymbol{x}' \in \mathcal{X} \setminus \mathcal{O}, q_{\text{Sp}}(\boldsymbol{x}) > q_{\text{Sp}}(\boldsymbol{x}') \big) \ge \max_{k \in \Lambda, \, \delta \in \Delta(k)} B(\gamma; \delta)$$

with $\Delta(k) = \{\delta \in \mathbb{R} \mid \mathcal{X}(\alpha; \delta) = \mathcal{O}\}$ if we set $\alpha = (n - k)/n$. Notice that $\gamma$ is determined from $\alpha$ (*i.e.* $k$) and $\delta$. Thus both $k$ and $\delta$ are implicitly optimized in $q_{\text{Sp}}$. In contrast, in $q_{k\text{thNN}}$ the number $k$ is specified by the user. For example, if $\Lambda$ is small, it is hardly possible to choose $k \in \Lambda$ without any prior knowledge, resulting in overlooking some outliers, while $q_{\text{Sp}}$ always has the possibility to detect them without knowing $\Lambda$ if $\mathcal{I}(\alpha; \delta)$ is non-empty for some $\alpha$. This difference in detection ability could be a reason why $q_{\text{Sp}}$ significantly outperforms $q_{k\text{thNN}}$ on average.

# 6   Conclusion

In this study, we have performed an extensive set of experiments to compare current distance-based outlier detection methods. We have observed that a surprisingly simple sampling-based approach, which we have newly proposed here, outperforms other state-of-the-art distance-based methods. Since the approach reached its best performance with small sample sizes, it achieves dramatic speedups compared to exhaustive methods and is faster than all state-of-the-art methods for distance-based outlier detection. We have also presented a theoretical analysis to understand why such a simple strategy works well and outperforms the popular approach based on $k$th-NN distances.

To summarize, our contribution is not only to overcome the scalability issue of the distance-based approach to outlier detection using the sampling strategy but also, to the best of our knowledge, to give the first thorough experimental comparison of a broad range of recently proposed distance-based outlier detection methods. We are optimistic that these results will contribute to the further improvement of outlier detection techniques.

**Acknowledgments.** M.S. is funded by the Alexander von Humboldt Foundation. The research of Professor Dr. Karsten Borgwardt was supported by the Alfried Krupp Prize for Young University Teachers of the Alfried Krupp von Bohlen und Halbach-Stiftung.

## Footnotes

[1]`http://sourceforge.net/projects/iforest/`

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
