[Supplementary Material · NIPS2013_0296_supplement.pdf]

# Supplementary Notes for
# Rapid Distance-Based Outlier Detection via Sampling

**Mahito Sugiyama**[1]   **Karsten M. Borgwardt**[1,2]
[1]Machine Learning and Computational Biology Research Group, MPIs Tübingen, Germany
[2]Zentrum für Bioinformatik, Eberhard Karls Universität Tübingen, Germany
{mahito.sugiyama,karsten.borgwardt}@tuebingen.mpg.de

## A   Definitions

Given a metric space $(\mathcal{M}, d)$ and a set of objects $\mathcal{X} \subset \mathcal{M}$, an object $\boldsymbol{x} \in \mathcal{X}$ is an *outlier* if

$$\big| \{ \boldsymbol{x}' \in \mathcal{X} \mid d(\boldsymbol{x}, \boldsymbol{x}') > \delta \} \big| \geq \alpha n,$$

where $n = |\mathcal{X}|$, the number of objects, and $\alpha$ and $\delta$ are parameters with $\alpha, \delta \in \mathbb{R}$ and $0 \leq \alpha \leq 1$. We denote the set of outliers by $\mathcal{X}(\alpha; \delta)$, which coincides with Knorr and Ng's $\mathrm{DB}(\alpha, \delta)$-outliers. Notice that $\alpha$ is usually large and close to 1 since outliers should be significantly different from almost all objects by definition. We also define

$$\overline{\mathcal{X}}(\alpha; \delta) := \mathcal{X} \setminus \mathcal{X}(\alpha; \delta)$$

and call an element in $\overline{\mathcal{X}}(\alpha; \delta)$ an *inlier*.

A $\delta$-*partition* $\boldsymbol{\mathcal{P}}_\delta$ of a set $\mathcal{X}$ is defined as a set of non-empty disjoint subsets of $\mathcal{X}$ such that each element (cluster) $\mathcal{C} \in \boldsymbol{\mathcal{P}}_\delta$ satisfies

$$\max_{\boldsymbol{x}, \boldsymbol{x}' \in \mathcal{C}} d(\boldsymbol{x}, \boldsymbol{x}') < \delta$$

and

$$\bigcup_{\mathcal{C} \in \boldsymbol{\mathcal{P}}_\delta} \mathcal{C} = \mathcal{X}.$$

We consider a $\delta$-partition of all inliers $\overline{\mathcal{X}}(\alpha; \delta)$ in Theorem 1 and 2, and that of the set $\mathcal{I}(\alpha; \delta) \subseteq \overline{\mathcal{X}}(\alpha; \delta)$ such that

$$\min_{\boldsymbol{x}' \in \mathcal{I}(\alpha; \delta)} d(\boldsymbol{x}, \boldsymbol{x}') > \delta$$

for all $\boldsymbol{x} \in \mathcal{X}(\alpha; \delta)$ in Theorem 3 and Corollary 1.

Our sampling-based method is defined as

$$q_{\mathrm{Sp}}(\boldsymbol{x}) := \min_{\boldsymbol{x}' \in S(\mathcal{X})} d(\boldsymbol{x}, \boldsymbol{x}')$$

using a randomly and independently sampled set $S(\mathcal{X}) \subset \mathcal{X}$. Thus our method requires as input only the sample size $s$ in practice, whereas the parameters $\delta$ and $\alpha$ are used only in our theoretical analysis.

Notation used in the paper is summarized in Table S1.

## B   Proof of Theorem 2

**Theorem 2** *Let $\boldsymbol{\mathcal{P}}_\delta = \{\mathcal{C}_1, \ldots, \mathcal{C}_l\}$ with $l$ clusters and $p_i = |\mathcal{C}_i| \,/\, n$ for each $i \in \{1, \ldots, l\}$. For every outlier $\boldsymbol{x} \in \mathcal{X}(\alpha; \delta)$ and the sample size $s \geq l$, we have*

$$\Pr\big( \forall \boldsymbol{x}' \in \overline{\mathcal{X}}(\alpha; \delta), \, q_{\mathrm{Sp}}(\boldsymbol{x}) > q_{\mathrm{Sp}}(\boldsymbol{x}') \big) \geq \alpha^s \sum_{\forall i; s_i \geq 0} f(s_1, \ldots, s_l; s, p_1, \ldots, p_l),$$

Table S1: Notation used in the paper.

| | |
|---|---|
| $\mathbb{R}$ | The set of real numbers |
| $d$ | distance function |
| $(\mathcal{M}, d)$ | Metric space |
| $\mathcal{X}$ | Set of objects; $\mathcal{X} \subset \mathcal{M}$ |
| $\boldsymbol{x}, \boldsymbol{x}', \boldsymbol{y}, \boldsymbol{y}'$ | Object, which is element of $\mathcal{X}$ ($m$-dimensional vector if $\mathcal{M}$ is multivariate) |
| $n$ | Number of objects, that is, $n = |\mathcal{X}|$ |
| $m$ | Number of dimensions |
| $S(\mathcal{X})$ | Sample set of $\mathcal{X}$; $S(\mathcal{X}) \subset \mathcal{X}$ |
| $s$ | Number of samples, that is, $s = |S(\mathcal{X})|$ |
| $\mathcal{X}(\alpha; \delta)$ | The set of DB$(\alpha, \delta)$-outliers of $\mathcal{X}$, that is, $\mathcal{X}(\alpha; \delta) = \{\, \boldsymbol{x} \in \mathcal{X} \mid |\{\boldsymbol{x}' \in \mathcal{X} \mid d(\boldsymbol{x}, \boldsymbol{x}') > \delta\}| \geq \alpha n \,\}$ |
| $\overline{\mathcal{X}}(\alpha; \delta)$ | The complement of DB$(\alpha, \delta)$-outliers, that is, $\overline{\mathcal{X}}(\alpha; \delta) = \mathcal{X} \setminus \mathcal{X}(\alpha; \delta)$ |
| $\boldsymbol{\mathcal{P}}_\delta$ | $\delta$-partition |
| $\mathcal{C}$ | Cluster (set of objects); $\mathcal{C} \in \boldsymbol{\mathcal{P}}_\delta$ and $\max_{\boldsymbol{x}, \boldsymbol{x}' \in \mathcal{C}} d(\boldsymbol{x}, \boldsymbol{x}') < \delta$ |
| $l$ | Number of clusters, that is, $l = |\boldsymbol{\mathcal{P}}_\delta|$ |
| $p_i$ | Fraction of $C_i$, that is, $p_i = |C_i| / |\bigcup_{\mathcal{C} \in \boldsymbol{\mathcal{P}}_\delta} \mathcal{C}|$ |
| $s_i$ | Possible outcome of number of samples in $C_i$, that is, $|C_i \cap S(\mathcal{X})|$ |
| $\mathcal{I}(\alpha; \delta)$ | Subset of $\overline{\mathcal{X}}(\alpha; \delta)$ satisfying $\min_{\boldsymbol{x}' \in \mathcal{I}(\alpha; \delta)} d(\boldsymbol{x}, \boldsymbol{x}') > \delta$ for all $\boldsymbol{x} \in \mathcal{X}(\alpha; \delta)$ |
| $\gamma$ | Fraction of $\mathcal{I}(\alpha; \delta)$, that is, $\gamma = |\mathcal{I}(\alpha; \delta)| / n$ |
| $f$ | The probability mass function of multinomial distribution |
| $\varphi$ | Function defined as $\varphi(s) := \sum_{\forall i; s_i \gtrless 0} f(s_1, \ldots, s_l; s, p_1, \ldots, p_l)$ |
| $B(\gamma; \delta)$ | Lower bound for $q_{\mathrm{Sp}}$ defined as $B(\gamma; \delta) := \gamma^s \max_{\boldsymbol{\mathcal{P}}_\delta} \varphi(s)$ |

*where $f$ is the probability mass function of the multinomial distribution defined as*

$$f(s_1, \ldots, s_l; s, p_1, \ldots, p_l) := \frac{s!}{\prod_{i=1}^{l} s_i!} \prod_{i=1}^{l} p_i^{s_i} \quad \text{with} \quad \sum_{i=1}^{l} s_i = s.$$

*Proof.* We have $\Pr(q_{\mathrm{Sp}}(\boldsymbol{x}) > \delta) = \alpha^s$ from the definition of outliers. Moreover, if $\mathcal{C}_i \cap S(\mathcal{X}) \neq \emptyset$, that is, $|\mathcal{C}_i \cap S(\mathcal{X})| \gtrless 0$ for all $i \in \{1, \ldots, l\}$, we have $q_{\mathrm{Sp}}(\boldsymbol{x}') < \delta$ for all $\boldsymbol{x}' \in \overline{\mathcal{X}}(\alpha; \delta)$. Such probability can be described using the probability mass function $f$ of the multinomial distribution:

$$\Pr\big(\forall \boldsymbol{x}' \in \overline{\mathcal{X}}(\alpha; \delta), q_{\mathrm{Sp}}(\boldsymbol{x}') < \delta\big) = \sum_{\forall i; s_i \geq 0} f(s_1, \ldots, s_l; s, p_1, \ldots, p_l),$$

where each $s_i \in \mathbb{N}$ corresponds to the outcome of the cardinality $|\mathcal{C}_i \cap S(\mathcal{X})|$ and the sum is taken over the following set

$$\left\{ (s_1, \ldots, s_l) \;\middle|\; \sum_{i=1}^{l} s_i = s \text{ and } s_i \gtrless 0 \text{ for all } i \in \{1, \ldots, l\} \right\}.$$

Thus the inequality follows. ∎

## C   Proof of Theorem 3

**Theorem 3** *Let $\boldsymbol{\mathcal{P}}_\delta = \{\mathcal{C}_1, \ldots, \mathcal{C}_l\}$ be a $\delta$-partition of $\mathcal{I}(\alpha; \delta)$ and $\gamma = |\mathcal{I}(\alpha; \delta)| / n$, and assume that $p_i = |\mathcal{C}_i| / |\mathcal{I}(\alpha; \delta)|$ for each $i \in \{1, \ldots, l\}$. For every $s \geq l$,*

$$\Pr\big(\forall \boldsymbol{x} \in \mathcal{X}(\alpha; \delta), \forall \boldsymbol{x}' \in \overline{\mathcal{X}}(\alpha; \delta), q_{\mathrm{Sp}}(\boldsymbol{x}) > q_{\mathrm{Sp}}(\boldsymbol{x}') \big) \geq \gamma^s \sum_{\forall i; s_i \gtrless 0} f(s_1, \ldots, s_l; s, p_1, \ldots, p_l).$$

*Proof.* If $S(\mathcal{X}) \subseteq \mathcal{I}(\alpha; \delta)$, then $q_{\mathrm{Sp}}(\boldsymbol{x}) > \delta$ holds for all $\boldsymbol{x} \in \mathcal{X}(\alpha; \delta)$, hence

$$\Pr\big(\forall \boldsymbol{x} \in \mathcal{X}(\alpha; \delta), q_{\mathrm{Sp}}(\boldsymbol{x}) > \delta\big) = \gamma^s.$$

In the same way as the above proof, if $\mathcal{C}_i \cap S(\mathcal{X}) \neq \emptyset$ for all $i \in \{1, \ldots, l\}$, we have $q_{\mathrm{Sp}}(\boldsymbol{x}') < \delta$ for all $\boldsymbol{x}' \in \overline{\mathcal{X}}(\alpha; \delta)$. The inequality therefore follows. ∎

## D Theoretical support for small sample sizes

**Remark** The lower bound $\alpha^s(1 - \beta^s)$ given in Theorem 1 with $0 < \beta < \alpha < 1$ is maximized at

$$s = \log_\beta \frac{\log \alpha}{\log \alpha + \log \beta}.$$

*Proof.* Let $g(s) = \alpha^s(1 - \beta^s)$. The differentiation of $g$ is obtained as follows.

$$\frac{dg}{ds} = \alpha^s \log \alpha - \left( (\alpha^s \log \alpha)\beta^s + \alpha^s(\beta^s \log \beta) \right)$$
$$= \alpha^s \left( \log \alpha - \beta^s(\log \alpha + \log \beta) \right).$$

Let us consider the behavior of the function

$$h(s) = \log \alpha - \beta^s(\log \alpha + \log \beta)$$

when $s$ increases from 0. Note that $\log \alpha < 0$ and $\log \beta < 0$ always hold since $0 < \beta < \alpha < 1$. It starts from a positive value since

$$h(0) = -\log \beta > 0$$

and becomes negative as we have

$$\log \alpha < \beta^s(\log \alpha + \log \beta),$$
$$h(s) = \log \alpha - \beta^s(\log \alpha + \log \beta) < 0$$

if $s$ is large enough. Moreover, this always holds when $s$ increases further since $\beta^s(\log \alpha + \log \beta)$ monotonically increases and

$$\lim_{s \to \infty} h(s) = \log \alpha < 0.$$

Thus $g$ takes the maximum value when $h(s) = 0$. It follows that

$$\beta^s(\log \alpha + \log \beta) = \log \alpha,$$
$$s = \log_\beta \frac{\log \alpha}{\log \alpha + \log \beta}. \qquad \blacksquare$$

Note that the sample size always takes a natural number, thereby technically we should check both the floor and ceiling and take the value which maximizes the bound $g(s) = \alpha^s(1 - \beta^s)$.

For intuitive understanding, we plot the sample size in Figure S1**a** which maximizes the lower bound $g(s)$ and the maximized lower bound in Figure S1**b** for $\alpha = 0.95, 0.99$, or $0.999$ with varying $\beta$ from 0 to 0.9. Such a large $\alpha$, which is close to 1, is a typical setting in outlier detection, as outliers should be significantly different from most of other objects by definition. As we can see, the probability of success is high and close to 1 for a wider range of $\beta$ if $\alpha$ is more and more close to 1. Moreover, The sample size is quite small and less than 50 in the presented cases, which is an attractive property of $q_{\text{Sp}}$ to achieve efficient outlier detection in massive data.

## E Comparison with $q_{k\text{thSp}}$

**Remark** For Wu and Jermaine's iterative sampling method $q_{k\text{thSp}}$, define

$$Z(\boldsymbol{x}, \boldsymbol{x}') := \Pr(q_{k\text{thSp}}(\boldsymbol{x}) > q_{k\text{thSp}}(\boldsymbol{x}'))$$

for an outlier $\boldsymbol{x} \in \mathcal{X}(\alpha; \delta)$ and an inlier $\boldsymbol{x} \in \overline{\mathcal{X}}(\alpha; \delta)$. We have

$$\Pr\left( \forall \boldsymbol{x} \in \mathcal{X}(\alpha; \delta), \forall \boldsymbol{x}' \in \overline{\mathcal{X}}(\alpha; \delta), q_{k\text{thSp}}(\boldsymbol{x}) > q_{k\text{thSp}}(\boldsymbol{x}') \right) \leq \min_{\boldsymbol{x} \in \mathcal{X}(\alpha; \delta)} \prod_{\boldsymbol{x}' \in \overline{\mathcal{X}}(\alpha; \delta)} Z(\boldsymbol{x}, \boldsymbol{x}').$$

*Proof.* Since each sampling is independent, if we focus on an outlier $\boldsymbol{x} \in \mathcal{X}(\alpha; \delta)$, we have

$$\Pr\left( \forall \boldsymbol{x}' \in \overline{\mathcal{X}}(\alpha; \delta), q_{k\text{thSp}}(\boldsymbol{x}) > q_{k\text{thSp}}(\boldsymbol{x}') \right) = \prod_{\boldsymbol{x}' \in \overline{\mathcal{X}}(\alpha; \delta)} Z(\boldsymbol{x}, \boldsymbol{x}').$$

As this holds for any outlier, the upper bound in the remark follows by considering all outliers. $\qquad \blacksquare$

Figure S1: The sample size (**a**) and the maximized lower bound (**b**) for $\alpha = 0.95$, $0.99$, or $0.999$ with varying $\beta$ from 0 to 0.9.

## F  Comparison with $q_{k\text{thNN}}$

**Remark**  Let $\mathcal{O} \subset \mathcal{X}$ be the set of true outliers given by an oracle and

$$\Lambda = \{\, k \in \mathbb{N} \mid q_{k\text{thNN}}(\boldsymbol{x}) > q_{k\text{thNN}}(\boldsymbol{x}') \text{ for all } \boldsymbol{x} \in \mathcal{O} \text{ and } \boldsymbol{x}' \in \mathcal{X} \setminus \mathcal{O} \,\},$$

which is the set of $k$s with which we can detect all outliers, and assume that $\Lambda \neq \emptyset$. Then we have

$$\Pr\big(\forall \boldsymbol{x} \in \mathcal{O}, \forall \boldsymbol{x}' \in \mathcal{X} \setminus \mathcal{O}, \, q_{\text{Sp}}(\boldsymbol{x}) > q_{\text{Sp}}(\boldsymbol{x}')\big) \geq \max_{k \in \Lambda, \, \delta \in \Delta(k)} B(\gamma; \delta)$$

if we set $\alpha = (n-k)/n$ and $\Delta(k) = \{\delta \in \mathbb{R} \mid \mathcal{X}(\alpha; \delta) = \mathcal{O}\}$.

*Proof.* Since $\mathcal{X}(\alpha; \delta) = \mathcal{O}$ for all $k \in \Lambda$ and $\delta \in \Delta(k)$, we have

$$\Pr\big(\forall \boldsymbol{x} \in \mathcal{O}, \forall \boldsymbol{x}' \in \mathcal{X} \setminus \mathcal{O}, \, q_{\text{Sp}}(\boldsymbol{x}) > q_{\text{Sp}}(\boldsymbol{x}')\big) \geq B(\gamma; \delta)$$

from Corollary 1. This inequality holds for all possible $k \in \Lambda$ and $\delta \in \Delta(k)$ simultaneously, and hence the remark follows.  ∎