[Reviews · NeurIPS 2013]

Submitted by Assigned_Reviewer_6

Summary:

This paper proposes a modification of the method of Wu and Jermanine [21]
for distance-based outlier detection. The empirical results show that
the proposed method outperforms [21] and in fact is generally the
best (and fastest) out of the 7 methods tested. To this reader it
is (at least at first) surprising that so few samples can be used
while retaining good performance.

Quality:

The modification over [21] is to only draw one sample from the
dataset, rather than repeated samples. This would seem to be an
effective strategy in terms of variance reduction. What is notable
from Fig 1 is that so few samples can be drawn in order to obtain good
performance (indeed it seems preferable to use a small number over n).

Table 2 shows that this is the fastest method, while table 3 shows
that method is only beaten on 4/20 datasets by [21].

In sec 4, please state what parameters (e.g. k) are being used here
for each method, and also why these are sensible choices.

Section 6 provides some analysis of why e.g. small sample sizes might be
desirable. I don't find this very compelling, e.g at line 394 they
talk about beta being small, but if there are several clusters
it seems that beta (defined ie eq 2) will be close to 1. To really
make this convincing I would like to see these analysis assumptions
tested against values obtained for real data.

Clarity: I found the writing clear

Originality: this could be seen as a simple "tweak" on [21], although a tweak
that appears to have a significant effect.

Significance: it appears that the proposed method advances the s-o-a
in distance-based outlier detection. Although this topic
is more "data mining" than the usual NIPS focus, it could be
of practical value.

Overall:

This paper proposes a simple modification of the method of Wu and Jermanine
[21] for distance-based outlier detection. The empirical results
presented show that the proposed method outperforms [21] and in fact
is generally the best (and fastest) out of the 7 methods tested. To
this reader it is (at least at first) surprising that so few sample
can be used while retaining good performance. The significance to the
NIPS community might be limited due to a relatively low profile
for outlier detection.


Summary: This paper proposes a modification of the method of Wu and Jermanine [21]
for distance-based outlier detection. The empirical results show that
the proposed method outperforms [21] and in fact is generally the
best (and fastest) out of the 7 methods tested. To this reader it
is (at least at first) surprising that so few samples can be used
while retaining good performance.

Submitted by Assigned_Reviewer_9

The authors propose a computationally efficient approach to outlier detection
which is compared with various prior schemes. Though potentially interesting,
the following issues emerged on reading the paper:

(1) In Section 3.3 and pen-ultimate paragraph, Section 1, there is
no description of the cutoff or decision for an outlier
over an inlier, which is obviously important. This issue is left to
Section 5 where a theorem is presented for the probability of an outlier
(the earlier description should give an indication of this later discussion).
The relevant material in Section 5 is not sufficiently self-contained,
in particular Knorr and Ng's DB(alpha,delta) needs to be described.
In the introduction to this Section, 'first we introduce a partition of
inliers into clusters using the threshold \delta', (a) how is this
done in practice, (b) how do we find the threshold in practice (what
happens if the data is all one cluster), (c) is there a model complexity
issue here (results depend on m, the number of clusters from Theorem 2).
Below (comment (3)) I suggest clearing some space in Section 3 so as to
fully describe how to derive a probability measure for an outlier and
the appropriate cutoff - perhaps with an illustrative numerical example.
The results in Section 4 do not give a clear enough descriprion
of the decision to label an datapoint as an outlier, given it is
left to later.

(2) Figure 1: says the x-axis has logarithmic scale but with number of
samples up to 1000, not sure what is meant here. Also, if the tests involve
data resampling then could add error bars.

(3) There is some unnecessary discussion of the alternative methods 3.1
onwards, e.g. kth-nearest neighbour for distance outlier detection is
known and could be more briefly described. As commented in (1) this
would free up space for more discussion of the probability of discriminating
an outlier over an inlier.

Development of efficient methods for outlier detection is an important issue with real-life applications and thus the paper could be interesting. As it stands I'm only recommending a marginal accept.
Summary: The authors propose a computationally simple sampling-based approach to outlier detection which they compare to various previous schemes for outlier detection. The paper is potentially interesting but I feel it has certain issues, outlined above.

Submitted by Assigned_Reviewer_11

Authors present a distance-based outlier detection method using a simple
but effective sampling-based approach. The new sampling-method
outperforms both in accuracy and in processing time as compared to a
variety of current state-of-art distance-based methods. The paper also
shows theoretical results proving why this approach finds the correct
outliers.

Pros:
- Achieves dramatic speed-ups compared to other algorithms due to effective sampling technique.
- Provides theoretical insights as to why the proposed algorithm is expected to have better performance.

Cons:
- Authors fix algorithm parameters independent of applied dataset to show performance results in Table 3. It would have been nice to compare performance results as reported by original papers of other algorithms (especially QkthNN, QkthSp). I think speed comparisons are valid. But claims on accuracy improvements needs further details.


How do distance-based approaches compare with model-based approaches
for outlier-detection in general? Any insights into where one works better
than others?
Summary: The proposed algorithm shows significant promise. Authors could have done a better job explaining parameter selection for their and comparable algorithms. In any case, this is well executed paper.
Author Feedback

Author rebuttal: Thank you for your time and effort and the positive feedback on the impact of our work. We provide answers to your questions below.


@Reviewer 11:

> Reviewer 11: Choice of algorithm parameters & Literature opinion on distance-based v.s. model-based approaches

> Answer: We performed q_kthNN [3, 5, 14, 15], q_kthSp [21], q_tree [14] and q_ABOF [16] with exactly the same parameter settings as in the original papers, q_LOF with the popular setting [14], and q_SVM with the popular heuristics [7] for a fair comparison.
In distance-based methods, hyperparameters of the algorithms have to be chosen by heuristics, as the unsupervised setting does not allow for cross-validation (similar to setting K in K-means, for example). Model (probabilistic) based approaches avoid such heuristics, but have been reported to still be outperformed by distance-based methods in practice (e.g. Bakar et al., A Comparative Study for Outlier Detection Techniques in Data Mining, 2006 IEEE Conference on Cybernetics and Intelligent Systems).


@Reviewer 6:

> Reviewer 6: “In sec 4, please state what parameters (e.g. k) are being used here for each method, and also why these are sensible choices.“

> Answer: We have described all parameter settings in Section 3. Since parameters of the algorithms have to be chosen by heuristics in distance-based methods, we have followed exactly the same settings as the original and subsequent papers for a fair comparison.

> Reviewer 6: Argument about Section 6, in particular about the appropriateness of the assumption about \beta.

> Answer: Regarding your example with many clusters and \beta being close to 1, it will make most (probably all) outlier detection algorithms fail, because clusters cannot be distinguished from groups of outliers in this setting.
Please note: \beta depends on \alpha and \delta and one has to set \delta large enough to get a realistic (not too large) number of outliers. A large choice of \delta will lead to a large size of clusters, which in turn leads to \beta being small. For instance, if we assume that 1% of our data are outliers and fix \alpha to be 0.95, then we have (\delta, the average of \beta) = (10.51, 0.50), (44.25, 0.0022), (10.93, 0.67), (37.10, 0.75), and (36.37, 0.80) on our first five datasets from Table 1 to achieve this 1% rate of outliers. We will follow your suggestion and include these numbers into the paper.


@Reviewer 9:

> Reviewer 9: Lack of description of the cutoff value of outliers in Section 1, 3.3, and 4

> Answer: Our algorithm returns a ranking according to outlierness, not a binary decision per point. In general, all distance-based outlier detection approaches merely rank the points according to their outlierness. A hard binary classification into outliers and inliers requires in all approaches to either know the fraction of outliers or a cut-off value beforehand. Hence our results compare rankings of outlierness in AUPRC plots (for all possible cut-offs). The theoretical analysis in Section 5 defines a cut-off value to then be able to make statements about the outlier recovery ability of our algorithm; but one does not need to set this cut-off value in practice. Please see also the next answer.

> Reviewer 9: “In the introduction to this Section, 'first we introduce a partition of inliers into clusters using the threshold \delta', (a) how is this done in practice, (b) how do we find the threshold in practice (what happens if the data is all one cluster), (c) is there a model complexity issue here (results depend on m, the number of clusters from Theorem 2). “

> Answer: (a, b) Our algorithm requires as input only the sample size \mu (as described in Section 3.3) and returns as output a ranking of all points according to their outlierness. \delta, \alpha and \beta are just parameters defined in the theoretical analysis to describe the probability that our algorithm will detect outliers from a given dataset with these parameters. They do not have to be set in practice. (c) There is no model complexity issue regarding m, as m should be minimized to achieve the optimal bound (the simplest clustering is the best) and, moreover, it is automatically chosen in Corollary 1.

> Reviewer 9: “The relevant material in Section 5 is not sufficiently self-contained, in particular Knorr and Ng's DB(alpha,delta) needs to be described. “

> Answer: In fact, DB(alpha, delta) is defined in both lines 114 and 344.

> Reviewer 9: Logarithmic scale of Figure 1 and error bars.

> Answer: Yes, the plot is really on logarithmic scale, as it is hard to see the important changes of AUPRC at small sample sizes (5 – 30) in a linear scale plot. We removed error bars since they indicate not the deviation of resampling but that among datasets, which will be large compared to changes in sample sizes and is confusing.